# Finding Stability—A Case Report on the Benefits of Adapted Kata Training for Children with Autism Spectrum Disorder

**DOI:** 10.3390/children11050523

**Published:** 2024-04-26

**Authors:** Bekir Erhan Orhan, Dilek Uzunçayır, Umut Canlı, Aydın Karaçam, Ali Selman Özdemir, Cristian Popa, Teodora-Mihaela Iconomescu, Laurențiu-Gabriel Talaghir

**Affiliations:** 1Faculty of Sport Science, Istanbul Aydın University, 34295 Istanbul, Türkiye; 2Institute of Educational Sciences, Anadolu University, 26470 Eskisehir, Türkiye; dilekuzuncayir@anadolu.edu.tr; 3School of Physical Education and Sports, Tekirdag Namik Kemal University, 59030 Tekirdag, Türkiye; ucanli@nku.edu.tr; 4Faculty of Sports Sciences, Bandırma Onyedi Eylül University, 10250 Bandırma, Türkiye; akaracam@dandirma.edu.tr; 5Faculty of Sports Sciences, Istanbul Topkapı University, 34087 Istanbul, Türkiye; aliselmanozdemir@topkapi.edu.tr; 6Faculty of Physical Educațion and Sport, Ovidius University from Constanța, 900527 Constanta, Romania; popa.cristian@univ-ovidius.ro; 7Faculty of Physical Educațion and Sport, Dunarea de Jos University from Galați, 800201 Galati, Romania; gtalaghir@ugal.ro

**Keywords:** adapted physical activity, karate, special needs education

## Abstract

This study investigated the efficacy of an Adapted Kata Training Program (AKTP) in enhancing balance for a 10-year-old child with Autism Spectrum Disorder (ASD), employing a mixed-model approach for data collection. Over 12 weeks, the AKTP demonstrated significant improvements in the child’s balance abilities, with an 11% increase in static balance, 8% in proprioceptive, 12% in horizontal, and 14% in vertical balance performance. These improvements persisted in a follow-up assessment after four weeks. Observations by the child’s mother corroborated the above findings. Consequently, this research suggests the AKTP as a valuable non-pharmacological intervention to improve balance in children with ASD. However, further studies are necessary to validate these results and explore the impact on additional developmental domains, such as cognitive and motor skills.

## 1. Introduction

Autism Spectrum Disorder (ASD) is a neurodevelopmental disorder with a neurobiological basis, presenting challenges such as speech issues due to social reluctance and learning difficulties that hinder effective communication [1]. These challenges often lead to communication breakdowns and an unwillingness to interact socially [1,2,3,4,5]. Repetitive behaviors and a lack of interest in new experiences can further impair social engagement and motor function, which is often exacerbated by muscle weakness in those participants not active in physical activities [1,2,3].

Gross motor skills, object control, and particularly balance are areas where individuals with ASD may experience difficulties [2,3,4,5,6,7,8,9,10,11]. Adequate balance is fundamental for basic motor skills and crucial for daily physical activity participation [12]. As a central component of motor functioning, balance involves maintaining a stable position, whether static or dynamic. The central nervous system and proprioception—the sensory feedback system for body position and movement—heavily influence it [13]. This sensory modality is vital for creating an internal body map [7,8,9,10,11,12,13,14].

Children with ASD frequently exhibit balance problems, affecting their ability to initiate, maintain, or end movements. These balance issues can also influence posture, often making individuals with ASD less balanced than their neurotypical peers [15,16,17]. The vestibular system coordinates posture and body movements and is crucial for balance [18]. However, sensory reception and modulation impairments within this system can lead to a refusal of movement, movement intolerance, and poor sensory registration in children with ASD [4,7,8,9,10,11,12,14,17,18,19]. Developing the vestibular system and its proper functioning can improve balance, motion perception, environmental recognition, and various skills, including social, auditory, and visual attention [19,20,21,22].

Motor symptoms such as tiptoeing can reduce balance and stamina in children with ASD, increasing the likelihood of falls and leading to more significant fatigue and less participation in social activities, which can negatively impact daily activities such as play, sports, and walking [7,8,9,10,11,12,13,14]. Although traditional treatments have provided moderate improvements in ASD-related behaviors, exercise and physical activities are being increasingly recognized as viable alternative treatments [23,24,25,26,27,28]. These treatments aim to enhance balance by stimulating the sensory systems, including the visual, vestibular, and proprioceptive systems, which have successfully improved joint proprioception and overall motor performance [28,29,30,31,32].

Recent research has highlighted the success of karate-based exercise programs in improving motor abilities in children with ASD, requiring skills such as speed, agility, muscle strength, flexibility, coordination, and balance [26,27,28,29,30]. Karate practice has also been suggested to be a powerful stimulus for neurological development in balance control [31]. Given the emerging evidence on the benefits of balance training for individuals with ASD and the variability of results across studies, there is a clear need for further research in this area [7,11,17,32]. This study, therefore, aims to address the gap in research involving children with special needs and examines the effects of an AKTP on balance in a 10-year-old diagnosed with ASD within the context of Türkiye, where there is limited research on supporting special educational needs [18,23,24,25,26,27,28].

## 2. Materials and Methods

This research utilized a convergent parallel mixed-methods design, integrating qualitative and quantitative data to evaluate the intervention’s effectiveness comprehensively. Creswell and Creswell’s [33,34] mixed-method research model facilitated simultaneous or sequential data collection and analysis, allowing for a more robust understanding and explanation of the observed phenomena. The study’s experimental design, a well-regarded quantitative method, examined the physical development of the child with ASD, ensuring no additional physical activities or classes were undertaken during the research period for accuracy [33,34].

### 2.1. Participant

The research participant, diagnosed with Autism Spectrum Disorder (ASD) following The Diagnostic and Statistical Manual of Mental Disorders, Fifth Edition (DSM-5) criteria [1], was a 10-year-old boy of 10 years and five months, an inclusion student alongside his peers, with a height of 164 cm and a weight of 66 kg. The parent was a 35-year-old high school graduate who is a full-time caregiver and homemaker. The study was limited to an individual diagnosed with ASD.

Participant prerequisites:Official ASD diagnosis;Does not exhibit behavioral problems (The information that the participant did not have any behavioral problems was obtained as a result of the participant’s health report and the information of his family.);No health problems;Not disturbed by social interactions (touching);Had not previously been enrolled in any physical activity programs or classes to enhance his balance skills for any objective.

The ASD participant was selected according to selective sampling as an ASD individual facing balance deficits. However, individuals with ASD vary considerably, and it is difficult to form a homogeneous group [35].

### 2.2. Measurement Tools

Balance was assessed using a moveable platform (Sensbalance MiniBoard; Sensamove^®^, Utrecht, The Netherlands), which provides an interactive training tool. While on the balance board, the participant was required to bring or hold the ball in the position required by the measured balance parameter, as seen on the screen, for 60 s. He was examined three times while completing two balance tasks under two conditions. In the measurements, a participant used a MiniBoard and a ball to perform two balance tasks. The first task required the participant to hold the ball steady at a specific position shown on the screen; during this, the MiniBoard could tilt approximately 10° in all directions. In the second task, the participant was expected to move the ball to a new specified position while maintaining balance on the MiniBoard. The MiniBoard is a sturdy wooden equipment designed to enhance the complexity of balance challenges and is capable of multi-directional or bi-directional tilting. First, the device uses innovative, non-invasive technology for real-time data recording. It offers storage in the form of Notepad data files, Excel, and graphical files. The apparatus allowed for tests on static and dynamic balance and ankle joint mobility [36]. This research focused on the subjects’ static and proprioception balance test measurements. In Figure 1, the visual of the measurement tool is given.

A recent study by Klostermann [37], assessing the reliability of Neuromuscular Control (NMC) tests with the use of Sensbalance MiniBoard and interactive courseware, suggests that the Sensbalance MiniBoard and interactive courseware can be used as an objective assessment tool for balance skills, neuromuscular control, proprioception, and motivation when supported by other tools. Future research should establish appropriate criteria for reliable use of this new tool.

### 2.3. Procedure

This study employed a pre-test, post-test, and retention test design to evaluate the influence of adapted kata training on the balance skills of a child with ASD. Before commencing the study, comprehensive information regarding the study’s purpose, objectives, anticipated outcomes, and research protocol was shared with the participant’s family, and their consent was obtained. Initial assessments, including a pre-test to gauge physical parameters and baseline balance, and a parent interview, were conducted before the 12-week training program. The participant then underwent 60 min of AKTP three times a week for the duration of the program. Upon completion, post-test measurements and a subsequent parent interview provided insight into the training’s impact on the child’s balance skills. After a four-week interval, retention tests were administered to evaluate the persistence of balance improvements. These structured interviews with the mother, conducted before the training program and after the retention period, each spanning approximately 30 min, complemented the quantitative data and enriched the study’s findings, as depicted in Figure 2.

Before taking measurements, the ASD individual played games for 30 min using the measurement tool to make it feel familiar, and their favorite toys and foods were used as reinforcement. Then, a trial was repeated three times for each measurement, and the highest measurement was taken after these trials.

Quantitative data were measured using the Sensamove Maxiboard balance device and then transferred to the SPSS 25.0 program, and statistical analyses were made. The data obtained were presented as tables and figures, and the differences between the pre-test, post-test, and retention test were determined. Demographic information about the child with ASD and their parent, and the views of the parent on the effects of the AKTP on the child, were collected through open-ended questions; qualitative interviewing is one data collection tool used to determine the experiences, knowledge, attitudes, and feelings of individuals [38,39,40]. The parent was interviewed before the AKTP started, the final interview was conducted 12 weeks later, and the retention interviews were four weeks after the end of the AKTP. All interviews were audio-recorded, and every effort was made to make the parent feel comfortable establishing honest feedback. In addition, the entire research process was recorded using the observation form created by researchers. Quantitative and qualitative data obtained were evaluated together. In qualitative data analysis, semi-structured interview forms were evaluated by making descriptive analyses, and the obtained quantitative and qualitative data were interpreted together.

A semi-structured interview format was utilized to collect data, focusing on the parents’ insights regarding the AKTP. The interviews aimed to capture the mother’s subjective experiences and opinions on the program’s effects. Each session was carefully documented through voice recordings to ensure the accuracy and richness of the qualitative data. This approach underlines the significance of parental perspective in evaluating intervention outcomes, supporting the need for comprehensive research that integrates such qualitative assessments into the broader understanding of the AKTP’s impact.

### 2.4. Data Analysis

The audio recordings of the interviews were translated into plain text without changes and then analyzed using a content analysis technique. The primary purpose of content analysis is to reach concepts and make connections that can explain the collected data. Themes were formed in line with common points from the answers given by the parent (mother) participating in the research, and standard features were compared and categorized. Additionally, to ensure validity, importance was given to reporting the collected data in detail and explaining how the researcher reached the results. The consensus coefficient between experts was calculated for the reliability of the research. The points where there was disagreement were discussed by both experts, and a consensus was reached. In order to ensure the reliability of the data, Miles and Huberman’s [41] formula (consensus/(consensus + dissent) × 100) was used. For the reliability of the research results, the consensus among experts was calculated as 96%.

### 2.5. Validity and Reliability

In qualitative research, providing a thorough report of the collected data and elucidating the steps taken to arrive at the results are imperative for ensuring validity [42]. Two essential procedures were employed to ensure the rigor and accuracy of the study. Firstly, a detailed description of the data analysis process was provided, allowing for transparency and clarity in the methodology. Secondly, quantitative and qualitative findings were presented collectively, enabling a comprehensive and nuanced understanding of the collected data. These measures were implemented to uphold the validity and reliability of the study. The questions in the interview form were submitted for review by expert academicians who have research in the field. After the necessary corrections were made, the questions were finalized, and three pilot interviews were conducted. The reliability of the research was calculated using a formula from Miles and Huberman [41] (trust = consensus/consensus + disagreement) by determining the number of times a consensus was reached and the number of disagreements. The result of the reliability calculation for the research is 96%. According to Miles and Huberman [41], a study is considered reliable if there is a 90% or more consensus among the researchers and experts in qualitative research. Accordingly, the study is reliable.

### 2.6. AKTP

The participant followed the Taikyoku Nidan Shotokan Karate Kata, a basic kata often taught to beginners. Each movement has its meaning and function. It consists of techniques for defense and attack in any direction at explosive speed against an imaginary opponent, moving in various directions in space following a specific order [43].

In implementing and planning the kata training program, the studies of Bahrami et al. [44] and Ansari et al. [45] were used. In the first two weeks of the study, sessions lasting 60 min involved only the teacher and the child. Kata-related games were played during these sessions to enhance communication, teach Kata techniques, and build the child’s confidence. These two sessions are part of the 12-week program. From the third week onwards, the session lasted 60 min, including a 10 min warm-up (running and stretching), 45 min of basic training (Taikyoku Jodan Techniques exercises), and a 5 min cool-down.

#### Kata

Kata has always been an integral part of karate practice. Kata is a series of moves designed to counter attacks from 360°, which means imaginary combat to neutralize an opponent. There are over 100 kata. The difficulty of these kata increases according to the kata group; examples of two groups, Gedan Barai and Oi Zuki, can be seen in Figure 3.

Every action in kata has its meaning and purpose. When drawing kata, karateka should imagine themselves surrounded by opponents and be ready to practice defensive and offensive techniques. A karateka starts to develop strength, speed, balance, and endurance by applying the kata formed from simple to complicated movements from the first block, punch, kick, knee, elbow strikes, bending, pushing, and pulling movements that they learned when starting the sport; kata unites the mind and body in a single discipline [43]. The research participant followed the kata shown in Figure 3.

Kata aid in developing dynamic balance as they target horizontal and vertical planes through different movements within one sequence, as seen below in Figure 4.

## 3. Results

Table 1 presents the percentage scores of a participant’s balance measurements over three distinct intervals. The static balance performance, which gauges the ability to remain stable at rest, shows an improvement from an initial 75% to 81% in the second measurement and further increases to 86% in the final measurement. Proprioceptive balance, relating to the body’s awareness of movement and position, also steadily increases from 73% to 77% to 81%. Regarding dynamic balance, the left–right horizontal measurement indicates the ability to maintain balance while moving side to side; this improved from 72% initially to 77% and then to 84%. Lastly, the front–back vertical balance, which assesses stability when moving forward and backwards, significantly progressed from an initial 81% to 84%, reaching 95% in the last measurement. Overall, the table indicates consistent improvements across all types of balance performance throughout the measurements.

Table 2 presents static and proprioceptive balance; the participants are expected to be as static or swing as little as possible; that is, the participants are expected to sway close to 0. At this point, it was determined that there were significant improvements in the sway performance of the participants. In the left–right horizontal dynamic balance performance, it is expected that the sway values of the participants to the right and left will increase while the sway values to the front and back will decrease. In this study, it was determined that the right and left sway values of the participants increased; that is, they moved away from the degree 0, while the front and back sway values approached the degree 0. In the forward–backwards dynamic balance performance, it is expected that the sway values of the participants to the front and back will increase while the sway values to the left and right will decrease. The study determined that the participants’ front and back sway values increased; that is, they moved away from the degree 0, while the left and right sway values moved closer to the degree 0. The participant’s balance performances are shown in Figure 5, Figure 6, Figure 7 and Figure 8.

Table 3 presents qualitative data from an interview focusing on a mother’s perspective regarding her child’s development across various dimensions—socialization, physical changes, psychological changes, and emotional changes—after the child’s participation in the AKTP.

Socialization: Initially, the child faced challenges in socializing, characterized by a lack of friends and communication barriers that contributed to sadness and social withdrawal. Over time, as the child engaged more in group activities and games through the AKTP, notable improvements were observed. The child developed confidence and happiness, indicating a positive shift toward active participation in social contexts. Challenges like physical limitations in walking and running initially hindered social interactions but improved as the child gained physical capabilities.

Physical Changes: Significant advancements were reported in the child’s physical condition. Before the intervention, the child required support for basic movements such as climbing stairs. Post-intervention, the child could navigate stairs independently, and issues like frequent stumbling and swaying were markedly reduced. These physical improvements were substantial enough to be recognized clinically, suggesting a robust impact of the AKTP on the child’s motor skills and overall physical autonomy.

Psychological Changes: Psychologically, the child exhibited reluctance and low motivation in new or challenging situations pre-intervention. However, post-AKTP, there was a notable enhancement in the child’s willingness to engage in activities, including increased participation in school. The program has fostered a sense of competence and self-efficacy, leading to greater independence and self-expression.

Emotional Changes: Before undergoing the AKTP, the child displayed significant distress responses, such as shouting and physical aggression, when faced with difficulties. Post-intervention, there was a notable improvement in emotional regulation, communication, and attention span. Additionally, the child’s ability to connect with peers evolved positively, moving from brief and selective interactions to more extended and meaningful engagements, which included improved verbal communication and eye contact.

## 4. Discussion

As a result of the 12-week kata training, the participant, a 10-year-old boy diagnosed with ASD, showed improvements in static and proprioceptive balance performance and dynamic balance performance measured in degrees on the horizontal and vertical planes. In addition, as a result of the third measurement, in which the retention of the detected developments was tested, it was determined that the parameters had been maintained. The parent had observed this in the participant’s ability to climb stairs and maintain balance in authentic spaces such as the playground.

Improvements in static and proprioceptive balance align with the results from Stins and Emck [10], who discuss the importance of static balance in the overall balance performance of individuals with ASD and suggest that interventions can improve static balance control. The gains observed in horizontal and vertical dynamic balance are consistent with Djordjević et al. [26] and Ji et al. [27], who reported that exercise interventions could positively impact dynamic balance in ASD, leading to improvements in motor skills that are crucial for daily activities.

Notably, the significant enhancement in front–back vertical balance from 81% to 95% indicates an improvement in the participant’s ability to control balance while moving forward and backward, which is essential for walking and other locomotor activities. This substantial increase is noteworthy as it surpasses improvements seen in other studies, suggesting that the targeted nature of the AKTP may be particularly beneficial for improving this balance component. Including the participant’s mother’s observations provides a qualitative dimension to the quantitative data, emphasizing the real-life applicability of the improvements noted in the clinical setting. Rosca et al. [24] highlight the role of parents’ observations in identifying and validating changes in a child’s balance and general motor abilities, arguing that parental insights can be a valuable source of information alongside objective measurements. It is also imperative to consider the study’s limitations when interpreting the results. The involvement of only one participant limits the generalizability of the findings, and Oster and Zhou [18] underline the need for larger-scale studies to validate the effectiveness of balance interventions in pediatric ASD populations. Moreover, the postural balance improvements identified in this study must be viewed cautiously, as they reflect short-term effects. The study by Hariri et al. [25] underlines the importance of examining the retention of these skills over an extended period to ensure the long-term efficacy of such interventions.

Balance levels are observed by standardized clinical tests in children [46,47,48,49,50]. It has been determined that children with ASD consistently score lower on such standardized tests compared to other children in the control group [4,46,47,48,49,50] and even have lower scores when compared with children with other types of (neuro)psychiatric disorders [39]. Despite this, recent studies show that physical activity and exercise practices are beneficial in developing balance and motor skills in children with ASD [30,44,45,51]. For example, swimming-based exercise programs are recognized as safe recreational activities for children with disabilities, including ASD [52,53,54]. Yilmaz et al. [55] reported that swimming could significantly improve static balance in children with ASD. We can define kata applications as an alternative exercise program based on different inputs, such as visual and somatosensory signals [56]. At this point, considering the importance of balance performance on daily functions, communication, and ability to interact, it is thought that karate kata may positively affect several of these areas for children with ASD [45] and is more accessible as no equipment is needed. When interviewed, the participant’s parent observed an increased willingness to socialize and communicate with peers, and the participant showed improvement in physical, social, psychological, and emotional well-being. The observations made by this research project then clearly support the ideas put forward by Kim et al. [57], who investigated the effects of an 8-week taekwondo training program on balance in 14 children with ASD. After eight weeks, they found that the taekwondo group showed a more significant improvement in one-leg stance balance than the control group. When other training protocols were examined, Battaglia et al. [58] determined that the swim-based program was adequate for developing gross motor skills (including balance) in adolescents with ASD. Sarabzadeh et al. [59] also revealed that six weeks of Tai Chi Chuan training could improve balance and movement coordination in children aged 6–12 with ASD.

It has been reported that static or dynamic balance is the primary variable for daily performances and motor performance in all children [56]. It is known that the number of studies evaluating the effect of various physical activity programs on improving balance in the ASD population has increased in recent years due to the effect of motor problems and impaired balance on daily performance, communication, and interaction skills [48,55,57,60,61]. Studies show that martial arts, another common form of physical activity, has positive physical and psychological effects on children with ASD [58,62,63]. In research from Ansari et al. [45], thirty students between the ages of 8 and 14 diagnosed with ASD were divided into three groups: kata training group, water exercise group, and control group. It was determined that there was a remarkable improvement in static and dynamic balance thanks to the training in kata techniques.

Interestingly, the karate group showed a more significant improvement in balance performance than the water exercise and control groups. This result indirectly affects karate training by improving body balance through specific movements and correct body alignment. Kata technique exercises such as kicking and blocking can ensure proper alignment of the feet, knees, hips, and spine, significantly improving proprioception and visual inputs as balance components. The whole interaction between neural and biomechanical mechanisms will provide an appropriate balance. The parts that trigger balance progression are a cooperation of postural muscle responses, better efficiency in vision, vestibular and somatosensory systems, adaptive systems, improved muscle strength, and better physical structure [59]. Therefore, the development of body awareness can be considered the most common cause of improved balance through martial arts [45].

## 5. Conclusions

In conclusion, this study explored the impact of a 12-week adapted kata training program on balance skills in a child with ASD. The training yielded significant improvements in the participant’s static, proprioceptive, and dynamic balance across both horizontal and vertical planes, suggesting the program’s efficacy in enhancing fundamental motor skills required for daily activities.

The notable persistence of balance improvements post-intervention, as evidenced by the retention test and corroborated by parental observations, underscores the potential of adapted kata training in promoting sustained physical development. This lasting effect indicates that the participant could integrate these skills into daily functions, an essential aspect of independence and quality of life for individuals with ASD.

However, while these findings are indicative, they call for a cautious interpretation due to the study’s limitations, such as the small sample size and lack of a control group. Future research involving larger cohorts and randomized controlled designs is necessary to substantiate the initial evidence presented. This will validate the effectiveness of kata training and potentially facilitate the generalization of these results to the broader ASD population. Moreover, subsequent research should consider investigating the impact of adapted kata training beyond balance skills to encompass cognitive and socio-emotional domains, offering a more comprehensive view of the benefits of physical activity programs for children with ASD.

The potential integration of adapted physical exercise programs into school curricula could revolutionize the support for students with ASD, providing an accessible, cost-effective method for enhancing balance and motor coordination. This study’s outcomes suggest that such programs could significantly contribute to the physical education framework, particularly benefiting children with special educational needs in Türkiye and potentially in broader contexts.

Overall, the promising results from this study advocate for the inclusion of martial-arts-based activities like kata training as a valuable component of therapeutic strategies aimed at addressing the unique challenges faced by children with ASD.

## Figures and Tables

**Figure 1 children-11-00523-f001:**
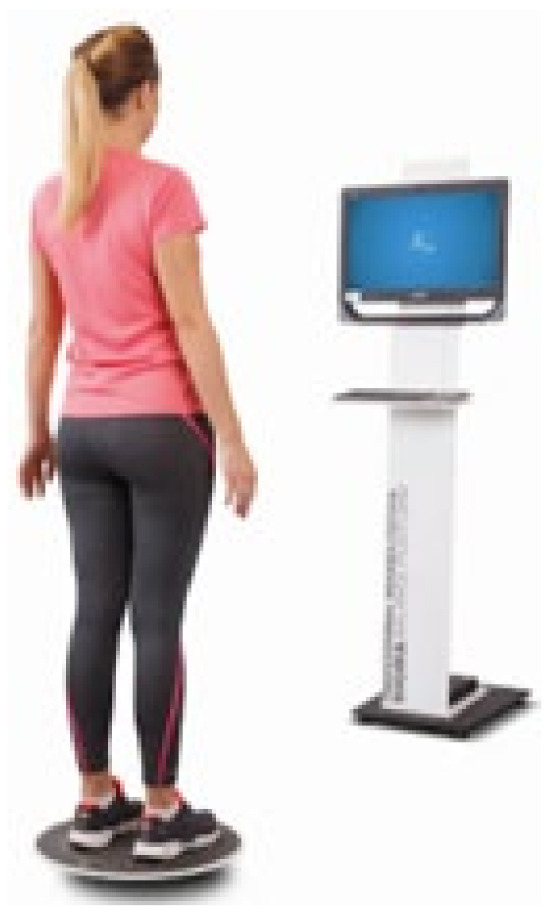
Measuring tool.

**Figure 2 children-11-00523-f002:**
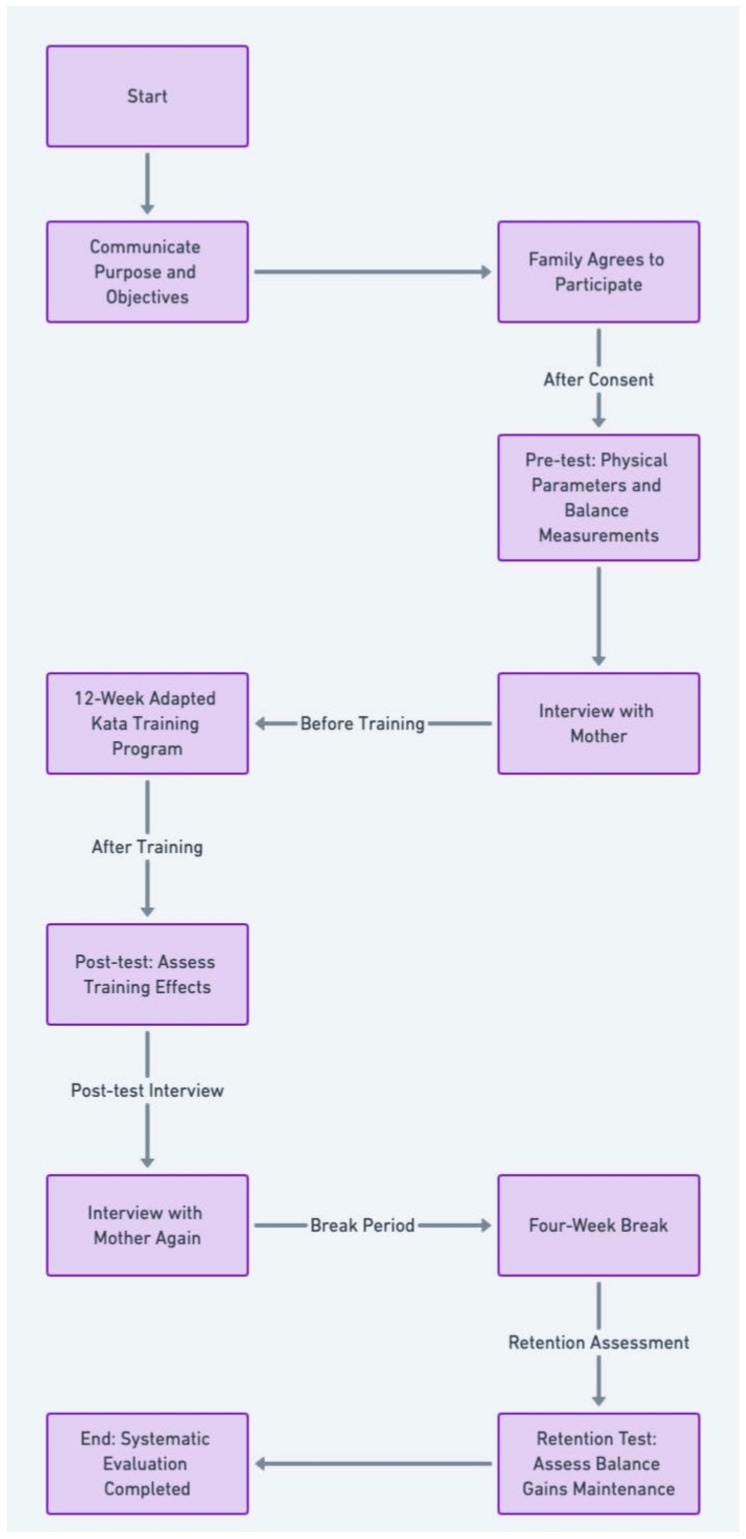
Procedure steps.

**Figure 3 children-11-00523-f003:**
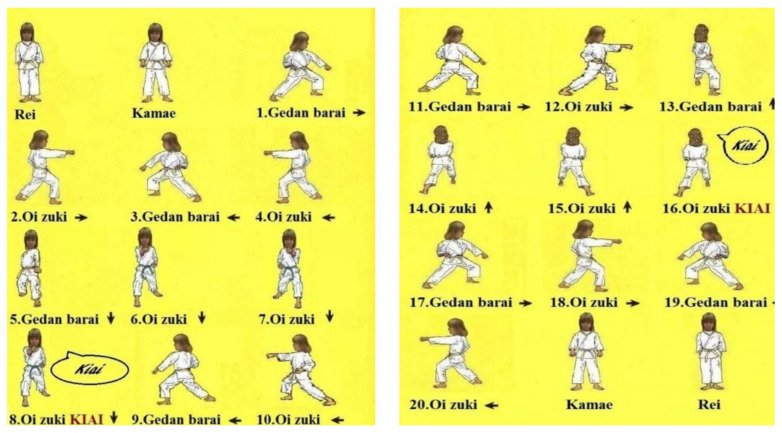
Application of Taikyoku Shodan kata techniques.

**Figure 4 children-11-00523-f004:**
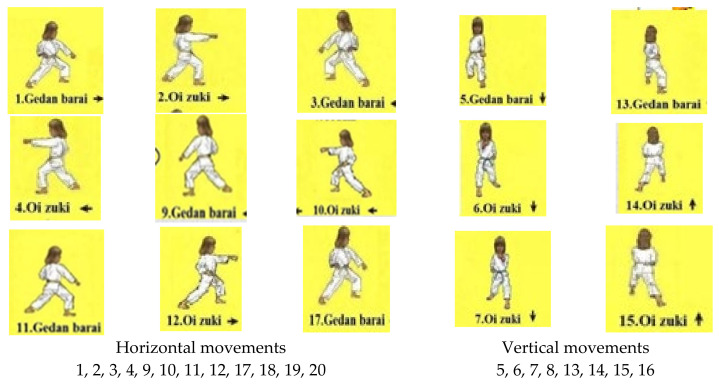
Horizontal and vertical movements in Taikyoku Shodan kata techniques.

**Figure 5 children-11-00523-f005:**

Figurative representation of three measurements of the participant’s static balance state in degrees (first and last measurements progress from **left** to **right**).

**Figure 6 children-11-00523-f006:**

Figurative representation of the three measurements of the participant’s proprioceptive balance state in degrees (first and last measurements proceed from **left** to **right**).

**Figure 7 children-11-00523-f007:**
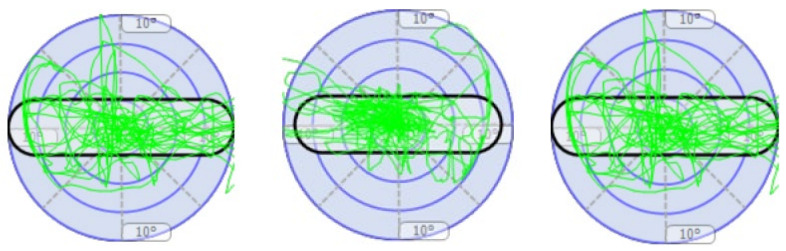
Figurative representation of the three measurements of the participant’s dynamic balance in degrees (left–right horizontal) (first and last measurements progress from **left** to **right**).

**Figure 8 children-11-00523-f008:**
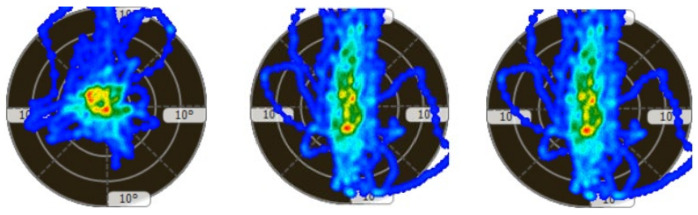
Figurative representation of the three measurements of the participant’s dynamic balance in degrees (front–back vertical) (first and last measurements progress from **left** to **right**).

**Table 1 children-11-00523-t001:** The percentage score for the participant’s balance measurement.

Balance Performance	1. Measurement (%)	2. Measurement (%)	3. Measurement (%)
Static	75	81	86
Proprioceptive	73	77	81
Left–Right Horizontal	72	77	84
Front–Back Vertical	81	84	95

**Table 2 children-11-00523-t002:** Sway distribution of balance parameters.

Average Deviation	Static Balance			Proprioceptive Balance			Left–Right Horizontal Dynamic Balance			Forward–Backwards Dynamic Balance		
	Measurements			Measurements			Measurements			Measurements		
	1.	2.	3.	1.	2.	3.	1.	2.	3.	1.	2.	3.
Front Avg. Dev.	0.85	1.69	1.01	0.37	0.40	0.39	1.91	1.20	1.02	2.37	2.42	2.46
Back Avg. Dev.	−1.74	−1.44	−0.79	−2.13	−1.66	−0.81	−1.29	−1.01	−0.90	−1.16	−1.20	−1.21
Left Avg. Dev.	−1.90	−0.86	−0.82	−1.78	−0.85	−0.65	−2.76	−2.82	−2.90	−1.75	−1.60	−1.50
Right Avg. Dev.	1.53	0.80	0.99	1.33	1.42	1.10	2.19	2.30	2.46	1.27	1.10	1.10

Avg. Dev: average deviation.

**Table 3 children-11-00523-t003:** Interview responses: mother’s view on the child’s development.

Categories	Question	Quotations
**Socialization**	**Did he have friends or spend time with other children regularly?** **What prevented him from socializing or hindered his socializing?**	*“He did not have many friends at school that he had communicated with before, and the situation made my son very sad; he did not want to talk too much and closed himself in communication. However, as the course went on, he spent more time with other kids, and as he got involved in the games and became successful, he became more confident and happier.”* *“We used to have many problems with walking and running; for these reasons, he was worried and closed himself off in social situations.”*
**Physical changes**	**Have you observed any change in his physical condition?** **What would you consider the most significant physical benefits for your son after following the programme?**	*“We noticed the change a lot; for example, before the AKTP, he could not climb the stairs without holding on and without support, but now he can go up and down the stairs independently.”* *“Before AKTP, they had much stumbling and falling almost every time he ran, and much swaying was in place. After AKTP, he occasionally falls like every normal kid, but it is almost non-existent, and his swaying has disappeared. Apart from the disappearance of the swaying, his body shape has started to improve. The doctor who followed my son noticed these developments and told us we should stop.”*
**Psychological changes**	**What was his typical reaction when going outside to play?** **How would you describe his confidence level before and after the program?**	*“When we went to the playground before AKTP, he quickly got bored and wanted to leave. I could see his fear and anxiety when on the playground. However, he has enjoyed the park very much and can play independently in the park.”* *“Previously, he was very reluctant to do something. I spent a long time convincing my son to do something. Usually, he was saying’’ I do not know that; I cannot do this. Later, he realized what he could and could do with the training, and now his motivation has increased in many places, including at school.”* *“Nowadays, he is expressing himself more with the self-confidence of acting completely independently.”*
**Emotional changes**	**What was a typical emotional response to difficulties or challenges?** **Was he able to build emotional connections with his peers?**	*“Before AKTP, my son had behaviours such as shouting, crying, throwing himself on the ground and hitting the person in front of him, albeit a little. However, he calmed down after the training, and his communication and attention span with us and his surroundings increased.”* *“While playing with his friends, he preferred to play for a short time and then get away from them, and he only spent time with children younger than him because he did not need to say much. However, after AKTP, he spoke a lot more. He started to trust himself, and even his eye contact improved; he seemed happier.”*

## Data Availability

The data presented in this study are available on request from the corresponding author.

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
