# Peer review of "Finding Stability—A Case Report on the Benefits of Adapted Kata Training for Children with Autism Spectrum Disorder"

_children, 2024, doi:10.3390/children11050523_

Round 1

Reviewer 1 Report

Comments and Suggestions for Authors

Dear authors,

I would like to congratulate you on the very interesting work you have done and thank you for allowing me to review it. I hope that the considerations I make will be useful to improve your manuscript. In the document attached, can find these considerations and suggestions.

Author Response

 Dear Prof.

We are very grateful for your valuable contributions and suggestions to the study. We have implemented the deficiencies and suggestions you mentioned in our research as best as possible.

The answers to your points are in italics in the file; you can see every change made to the manuscript.
Thank you again for your valuable contributions.

Best regards.

Reviewer 2 Report

Comments and Suggestions for Authors

Comments to the author:

Manuscript ID: children-2963378

Title: “Finding Stability - A Pilot Study on the Benefits of Adapted Kata Training for Children with Autism Spectrum Disorder”.

Thank you for giving me the opportunity to review this manuscript. The study investigated the effectiveness of a 12-week Adapted Kata Training Program (AKTP) in improving the balance skills of a 10-year-old child with Autism. Even though the theme of the study is interesting, I feel that the manuscript in its current form is unsuitable for publication, and major revisions are needed. My comments are listed below:

General comment

The introduction lacks appropriate citations in several lines of the body.

The material and methods section includes unnecessary theoretical information, and several important information are in the wrong place in the manuscript. A reorganization of this section to be more reader-friendly is vital.

The discussion contains findings from research which should be included first in the introduction section. Authors must present all relevant research findings in the introduction section first and then in the discussion should compare and discuss their findings in relationship with the findings from other studies.

Abstract

Lines 17-18. Please delete lines 17-18. They do not add significant information to the research.

Line 26. Add the word above before the word findings.

Lines 26-27. The statement you make in these lines is strong and it is not supported by the findings of your study since you examined only one child. Please, rephrase or delete the sentence.

Line 31. In what domains of development do you refer? Please, specify.

Introduction

Line 36-37. The citations are missing. Add citations.

Lines 37-38. I cannot understand what you mean when you write that children with ASD need more words due to learning difficulties. Please, rephrase the sentence.

Line 39. After the words “breakdowns” add a citation.

Line 41. Who states that participants with ASD have low motivation in undertaking new activities and engaging with peers? Please, add a citation.

Lines 48 and 50. Citations are missing.

Line 69. The citation is missing.

In the introduction, you should explain further the relationship between balance and fundamental movement skills, which are rather delayed or impaired in children with ASD. Also, you must refer that more than 50% of individuals with ASD are born with hypotonia. Moreover, the bibliography used is rather limited and outdated. In some cases is even irrelevant to the scope of your study e.g., “Bumin, G.; Uyanik, M.; Yilmaz, K.; Kayihan, H.; Topçu, M. Hydrotherapy for Rett syndrome, 2003”. Rett syndrome is no longer considered a type of autistic disorder, but a neurodegenerative disorder. Please, check the following citations:

Stins JF, Emck C. Balance Performance in Autism: A Brief Overview. Front Psychol. 2018 Jun 5;9:901. doi: 10.3389/fpsyg.2018.00901. PMID: 29922206; PMCID: PMC5996852.

RoÈ™ca, A.M.; Rusu, L.; Marin, M.I.; Ene Voiculescu, V.; Ene Voiculescu, C. Physical Activity Design for Balance Rehabilitation in Children with Autism Spectrum Disorder. Children 2022, 9, 1152. https://doi.org/10.3390/children9081152

Oster LM, Zhou G. Balance and Vestibular Deficits in Pediatric Patients with Autism Spectrum Disorder: An Underappreciated Clinical Aspect. Autism Res Treat. 2022 Aug 16;2022:7568572. doi: 10.1155/2022/7568572. PMID: 36016580; PMCID: PMC9398866.

Rabeeh Hariri, Amin Nakhostin-Ansari, Fatemeh Mohammadi, Amir Hossein Memari, Iman Menbari Oskouie, Afarin Haghparast, "An Overview of the Available Intervention Strategies for Postural Balance Control in Individuals with Autism Spectrum Disorder", Autism Research and Treatment, vol. 2022, Article ID 3639352, 9 pages, 2022. https://doi.org/10.1155/2022/3639352

Djordjević M, Memisevic H, Potic S, Djuric U. Exercise-Based Interventions Aimed at Improving Balance in Children with Autism Spectrum Disorder: A Meta-Analysis. Percept Mot Skills. 2022 Feb;129(1):90-119. doi: 10.1177/00315125211060231. Epub 2021 Dec 22. PMID: 34936828.

Ji YQ, Tian H, Zheng ZY, Ye ZY, Ye Q. Effectiveness of exercise intervention on improving fundamental motor skills in children with autism spectrum disorder: a systematic review and meta-analysis. Front Psychiatry. 2023 Jun 12;14:1132074. doi: 10.3389/fpsyt.2023.1132074. PMID: 37377477; PMCID: PMC10291092.

Lines 90-91. This statement is untrue. Please, rephrase.

Materials and Methods

Lines 98-112. Please delete the 2.1 subsection. There is no need to add all this information.

I propose you reorganize the whole section starting with 2.1 “Participants”, 2.2 “Measurement Tools”, 2.3 “Procedure” and 2.4 “Statistical analysis”

Lines 132-146. These lines should constitute the Statistical analysis section.

Lines 150-151. Please remove the phrase: “The parent and informed consent forms were completed before beginning the research” from the participants section to the procedure section.

Line 153. A formal diagnosis of ASD. Please clarify who gave the diagnosis and according to which diagnostic criteria (DSM-V/ ICD-11/ ICD-10). Also, clarify whether he attended a regular school setting or a special school and if he had normal intelligence.  

Lines 157-158. I think you want to write that he had not previously participated in any physical class or activity aiming to enhance his balance skills. 

Line 160. Name the exact age of the participant in age and months e.g. 9;10.

Line 169. Delete this line.

Line 177. Describe the tasks that the participant with ASD had to perform.

Lines 181-193. Please refer to only the important information for the reliability of the tool.

Line 194. Please delete this line.

Lines 204-205. Remove these lines from here and place them in the procedure section.

Lines 206-207. Delete this statement. Does not provide important information for the construction of the information form or the semi-structured interview.

Lines 226-230. These lines should be removed from here and placed after the 205 line. Delete all other information from this paragraph.

Results

Line 267. Please, clarify in the text that the 1st measurement is the pre-test measure, the second is the post-test measure and the 3rd measurement is the retention measure?

Discussion

Lines 303-310. All information referred to here should first be presented in the introduction section.

Line 311. Rett syndrome is no longer categorized within autism spectrum disorders. Please remove this study.

Research from training programs showing positive effects on balance skills in ASD children should be first presented in the introduction section and not in the discussion.

Comments on the Quality of English Language

-

Author Response

Dear Prof.

We sincerely thank you for your insightful and constructive feedback on our manuscript, "Finding Stability—A Case Report on the Benefits of Adapted Kata Training for Children with Autism Spectrum Disorder." Your thorough review and comprehensive comments have significantly enhanced our manuscript.

Your discerning eye for detail has improved our paper's clarity and flow and reinforced our research's academic rigour. Your guidance, particularly in organizing the Materials and Methods section and the critical review of our bibliography, was invaluable. It has allowed us to present our findings in a more precise and reader-friendly manner, thus elevating the scholarly value of our work.

We have carefully noted each point you raised and diligently worked to revise our manuscript accordingly. These modifications have undoubtedly strengthened our study's narrative and scientific validity.

Again, we appreciate the time and effort you dedicated to the review process. Your contributions were instrumental in refining our study, and we are deeply thankful for your role in bringing our research to its fullest potential.

Warm regards.

Reviewer 3 Report

Comments and Suggestions for Authors

The paper presented a case study on the effect of Kata training on balance and physical, psychological, and emotional changes in a ten-year-old boy with ASD. The study combines qualitative and quantitative research methods, i.e., interviews with a parent and measurements evaluating static and dynamic balance and proprioception sense. However, the measurements related to the static and dynamic balance and the sense of proprioception are mainly presented visually. Neither the magnitude of sway nor its standard deviation in static and dynamic conditions nor the proprioceptive balance are provided. Only the overall percentage score of the measurements in the initial, final, and retention phases, whereas the authors state that their equipment ensures the possibility of performing analysis on the data as the results are provided not only in graphical format but in tabular form. Moreover, the authors claimed the performance of statistical tests.

For example, it would be good to provide data on whether pose improvement is asymmetric in the vertical and horizontal direction, taking into account the dominance of the horizontal movements in the training set.

Thus, more quantitative data on the effect of the training needs to be provided.

Also, the authors selected a participant without behavioral problems for this case study but did not provide any information about how this was evaluated.

Minor issues:

Not all references follow the requirements of the journal.

Some of the labels in Table 3 are not in English.

Author Response

Dear Prof.

We are very grateful for your valuable contributions and suggestions to the study. We have implemented the deficiencies and suggestions you mentioned in our research as best as possible.

The answers to the points you mentioned are given in the file, and you can see every change made in the Manuscript.

Thank you again for your valuable contributions.

Best regards.

Round 2

Reviewer 1 Report

Comments and Suggestions for Authors

Dear authors,

The changes made to the document, I believe, have greatly improved the text. However, new issues have arisen, which require further revision.

You have the explanations in the attached document. 

Kind regards,

Author Response

Dear Prof.

We extend our profound gratitude for your astute observations and invaluable contributions to improving our manuscript. Your meticulous scrutiny and subsequent suggestions have refined the presentation of our results and imbued greater accuracy and clarity within our research.

The linguistic corrections you recommended were fundamental in ensuring that the manuscript adheres to the highest standards of academic English. As suggested, the unification of the data tables has significantly improved the coherence and succinctness of our findings, facilitating a more precise understanding for future readers. Moreover, your discerning analysis of the data interpretation has enabled us to convey our findings more precisely.

Your input has been instrumental in elevating the quality of our work and has undoubtedly enriched the manuscript. We sincerely appreciate the depth of expertise and the attention to detail you have brought to this revision process.

We believe that collaborative efforts such as yours greatly enhance scholarly discourse, leading to the collective advancement of our scientific understanding. Once again, we thank you for your exemplary contribution to refining our study.

Best regards.

Reviewer 2 Report

Comments and Suggestions for Authors

Manuscript ID: children-2963378

 Title: Finding Stability - A Pilot Study on the Benefits of Adapted Kata Training for Children with Autism Spectrum Disorder

 The revised version of the manuscript has addressed many issues of the previous submission. Thank you for your efforts. However, I still suggest some minor revisions, before the article is ready for publication. You will find my comments below:

 Line 129. Put only the abbreviation. You have written what it means AKTP upwards in your text.

Line 194. It is DSM-5, not DSM-V. Also, you must write it first in full and then put in bracket the abbreviation.

Line 201. Please, change the word official to formal and specify that the diagnosis was given according to DSM-5 criteria (APA, 2013).

Lines 111-112. Delete the phraseWhile selecting the individual with ASD participating in the study”. Then start with a capital letter and replace the phrase with the following: “The ASD participant was selected according to selective sampling as an ASD individual facing balance deficits.

Line 223. Refer to exactly the two balance tasks (exercises) and the two conditions e.g., the participant had to keep the miniBoard stabilized while maintaining body posture or move miniBoard front-back and keep his body posture. Also, refer to the equipment of the miniBoard used e.g. A sturdy rubber accessory allowing tilting of approx. 15Ëš in all directions or a sturdy rubber accessory allowing tilting of approx. 15Ëš in two directions (Front-Back, or Left-Right).

Lines 231-238. Please shorten this paragraph to include only important information e.g., "A recent study by Klostermann [37], assessing the reliability of Neuromuscular Control (NMC) tests with the use of Sensbalance MiniBoard and interactive courseware suggests that the Sensbalance MiniBoard and interactive courseware can be used as an objective assessment tool for balance skills, neuromuscular control, proprioception, and motivation when supported by other tools. Future …."

Line 289. Delete this subtitle “Statistical analysis” and combine the following paragraph with the procedure section.

The discussion is well-organized and well-written.

Author Response

Dear Prof.

We would like to express our profound gratitude for your invaluable guidance and insightful contributions throughout the development of our work. Your detailed and constructive feedback has been instrumental in enhancing the structure and content of each section. The depth of your expertise has significantly enriched our study, and your support has been indispensable to our progress.

Best regards.
